# Deep Learning Approach for Cardiac Electrophysiology Model Correction

**Victoriya Kashtanova** [1]  **Mihaela Pop** [1]  **Patrick Gallinari** [2]  **Maxime Sermesant** [1]

## Abstract

Imaging the electrical activity of the heart can be achieved with invasive catheterisation. However, the resulting data are sparse and noisy. Mathematical modelling of cardiac electrophysiology can help the analysis but solving the associated mathematical systems can become unfeasible. It is often computationally demanding, for instance when solving for different patient conditions. We present a new framework to model the dynamics of cardiac electrophysiology at lower cost. It is based on the integration of a low-fidelity physical model and a learning component implemented here via neural networks. The latter acts as a complement to the physical part, and handles all quantities and dynamics that the simplified physical model neglects. We demonstrate that this framework allows us to reproduce the complex dynamics of the transmembrane potential and to correctly identify the relevant physical parameters, even when only partial measurements are available. This combined model-based and data-driven approach could improve cardiac electrophysiological imaging and provide predictive tools.

## 1. Introduction

Despite the fact that biophysically detailed cardiac electrophysiology (EP) models (such as (Ten Tusscher et al., 2004)) can accurately reproduce electrical behaviour of cardiac cells, these models are complex and computationally expensive, and have many hidden variables which are impossible to measure, making model parameters difficult to personalise. The phenomenological models (FitzHugh, 1961; Nagumo et al., 1962; Aliev & Panfilov, 1996; Nash & Panfilov, 2004; Mitchell & Schaeffer, 2003), simplified models derived from biophysical models, have fewer parameters

and are therefore especially useful for rapid computational modelling of wave propagation at the organ level. However, they are less realistic and therefore need a complementary mechanism to fit them to the measured data. Machine learning and in particular deep learning (DL) approaches could help providing such a correction mechanism. The combination of rapid phenomenological models and machine learning components could then allow the development of rapid and accurate models of transmembrane dynamics (as in (Fresca et al., 2021; Sahli Costabal et al., 2020)). Nevertheless, the majority of existing coupled approaches bases on high-fidelity physical models and fits them to the data. This could be computationally expensive and cannot manage large discrepancies between simulated and real data.

To alleviate this limitation, we propose a framework to learn complex cardiac electrohysiology dynamics from data, based on a fast low-fidelity (or incomplete) physical model. This framework has two components which decompose the dynamics into a physical and a data-driven term. The data-driven deep learning component is designed so as to capture only the information that cannot be modeled by the incomplete physical model. The proposed model closely follows the approach of (Yin et al., 2021). But in contrast to this work, that considers fully-observable dynamics and simple test use cases, cardiac electrophysiology dynamics have a high complexity and represent simultaneously multiple underlying processes. Furthermore, most cardiac electrophysiology models lack measurements for some variables, which makes them partially-observable and requires inferring the dynamics from incomplete observations only. Fig. 1 presents the general framework of our approach. Training amounts to identifying the physical model parameter (inverse problem) and learning the neural network parameters (direct problem) together. After training, the model can be used for forecasting at multiple horizons.

## 2. Learning Framework

To learn cardiac electrophysiology dynamics $(X_t)$ we solve an optimisation problem via a physics-based data-driven *Our framework's name* framework. This framework combines a physical model $(F_p)$ representing an incomplete description of the underlying phenomenon and a neural network $(F_d)$ which will complement the physical model by

[1]Inria d'Université Côte d'Azur, Sophia Antipolis, France [2]Sorbonne University, Criteo AI Lab, Paris, France. Correspondence to: Victoriya Kashtanova <victoriya.kashtanova@inria.fr>, Maxime Sermesant <maxime.sermesant@inria.fr>.

*Workshop on Interpretable ML in Healthcare at International Conference on Machine Learning (ICML)*, Honolulu, Hawaii, USA. 2023. Copyright 2023 by the author(s).

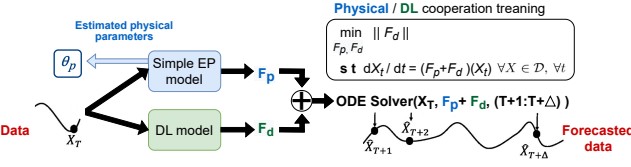

Figure 1. General *Our framework's name* framework scheme. During the training phase two-component framework learn the parameters for the physical ($F_p$) and the data-driven ($F_d$) components from data. Then via an ODE solver the framework can forecast further the learned dynamics.

capturing the information that cannot be modeled by the physics component:

$$\min_{F_p\in\mathcal{F}_p, F_d\in\mathcal{F}_d} \|F_d\| \text{ subject to}$$
$$\forall X \in \mathcal{D}, \forall t, \frac{dX_t}{dt} = F(X_t) = (F_p + F_d)(X_t). \quad (1)$$

Our incomplete physical model is the two-variable $(v, h)$ model (Mitchell & Schaeffer, 2003) for cardiac EP simulations, as described by equations (2). The variable $v$ represents a normalised ($v \in [0, 1]$) dimensionless transmembrane potential, while the "gating" variable $h$ controls the repolarisation phase (i.e., the gradual return to the initial resting state):

$$\partial_t v = \frac{hv^2(1-v)}{\tau_{in}} - \frac{v}{\tau_{out}} + J_{stim}$$
$$\partial_t h = \begin{cases} \frac{1-h}{\tau_{open}} & \text{if } v < v_{gate} \\ \frac{-h}{\tau_{close}} & \text{if } v > v_{gate} \end{cases} \quad (2)$$

where $J_{stim}$ is a transmembrane potential activation function, which is equal to 1 during the time the stimulus is applied ($t_{stim}$).

This physical model has been successfully used in patient-specific modelling (Relan et al., 2011), covering general EP dynamics. Furthermore, in contrast to the very detailed ionic/cellular models, this model is flexible in terms of spatial and temporal steps set in the numerical analysis.

The data-driven component ($F_d$) of the framework was implemented via a neural network. The choice of a neural network depends on the application problem and the dimension of the data. In this work, we used a ResNet network (He et al., 2016), because it could accurately reproduce complex cardiac EP dynamics (Ayed et al., 2019; Kashtanova et al., 2021). However, a simpler neural network could also be used for more rapid computations.

Instead of solving the ODE in Eq. (1), we use an integral trajectory-based approach which is robust and less sensitive

**Algorithm 1** *Our framework's name* training

Initialisation: $\theta_0, \lambda_0 \geq 0, \tau > 0$;
**for** epoch = $1 : N_{\text{epochs}}$ **do**
  **for** batch in $1 : B$ **do**
    $\mathcal{L}_{\text{traj}}(\theta_j) = \sum_{i=1}^{N}\sum_{h=1}^{T/\Delta t} ||X_{h\Delta t}^{(i)} - \tilde{X}_{h\Delta t}^{(i)}||$
    $\theta_{j+1} = \theta_j - \nabla[\lambda_j \mathcal{L}_{\text{traj}}(\theta_j) + \|F_d\|]$
  **end for**
  $\lambda_{j+1} = \lambda_j + \tau \mathcal{L}_{\text{traj}}(\theta_{j+1})$
**end for**

to the time resolution (Yin et al., 2021). We compute the next state $\tilde{X}_{h\Delta t}^{(i)}$ from the initial state $X_0^{(i)}$ as an approximate solution of the integral $\int_{X_0^{(i)}}^{X_0^{(i)}+h\Delta t}(F_p^{\theta_p} + F_a^{\theta_a})(X_s)\,dX_s$ obtained by a differentiable ODE solver (Chen et al., 2018; 2021). The *Our framework's name* training uses an algorithm adapted from (Yin et al., 2021).

## 3. Experiments and Results

In our previous work, using in silico data, we demonstrated ability of the framework to reproduce the complex dynamics of transmembrane potential including a case where noise is present in the data.

Here, in order to test the performance of our *Our framework's name* framework and to further show its capability to reproduce transmembrane potential dynamics of different complexities, we performed a series of real data experiments. Using optical fluorescence imaging data of action potentials recorded ex vivo on explanted porcine hearts, we aimed to show that our framework can be easily personalised on real data and can identify key physical parameters for different anatomical zones having abnormal electrical function.

The details of data collection used for the experiments are presented in detail below.

### 3.1. Data collection

We tested *Our framework's name* framework performance on ex vivo datasets from optical fluorescence imaging of action potential. Briefly, the optical signals were recorded ex vivo on an heart explanted from a juvenile swine ( 25kg in weight). The heart was attached to a Langendorff perfusion system and a voltage sensitive dye (i.e., di-4-ANEPPS) was injected into the perfusate. In order to avoid cardiac motion artifacts during the electrophysiological recordings, the heart contraction was suppressed by a bolus of saline and Cytochalasin D, an electro-mechanical uncoupler. All optical images were acquired epicardially using a high-speed CCD camera (MICAM02, BrainVision Inc. Japan), with high-temporal resolution (3.7ms) as well as a high-spatial resolution (i.e., pixel size 0.7mm x 0.7mm). The action

potential was then derived at each pixel from the relative change in the intensity of fluorescence signal. The experiments are described in more detail in (Pop et al., 2009).

For this experiment, we used a heart with an ischaemic region. We manually selected two rectangular regions of interest (ROIs) with different action potential dynamics across time (see Figure 2). Next, we normalised the optical signal, to obtain a [0, 1] min/max interval for transmembrane potential, while keeping the noise in the data. We took a first full cardiac cycle and removed the parts with zero potential, keeping only time-sequences of 300 ms per experiment. Then, we saved a time sequence for each pixel from each ROI in separate files, creating two databases (ROI A and ROI B) containing each about 10 and 5 time-sequences for training and validation respectively.

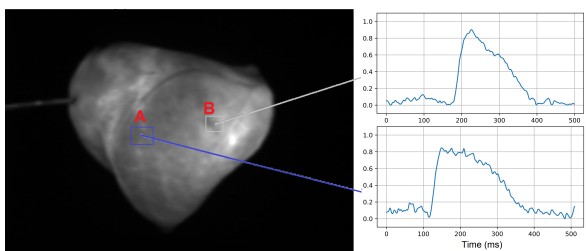

*Figure 2.* Example of optical mapping data (tracings of denoised action potential waves) recorded ex vivo in a porcine heart. ROI B represents an ischaemic region characterised by a shorten action potential duration (APD) compared to the normal APD recorded in ROI A.

The optical data were considered here as the ground truth. Our specific objective was to learn the complex dynamics of measured action potential, and then to identify the relevant physical parameters for different parts of the heart.

### 3.2. Results

Using optical imaging mapping data, our *Our framework's name* framework was able to reproduce the observed action potential dynamics for different ROIs within the heart, identifying the 3 major physical dynamics parameters ($\tau_{in}$, $\tau_{out}$ and $\tau_{close}$). Figures 3 and 4 demonstrate that the framework correctly estimated the difference in value for the parameter $\tau_{close}$, which either increased APD or shortened it, respectively.

Table 1 summarises the quantitative results for our framework forecasting on train and validation data samples, in comparison to baseline methods trained on the same data. To calculate this error, for each data sample, we fed the framework with only one initial measurement, then let it predict 300 ms forward without any additional information.

The obtained MSE is relatively small for both ROI, and, despite the use of a limited dataset for training, the *Our*

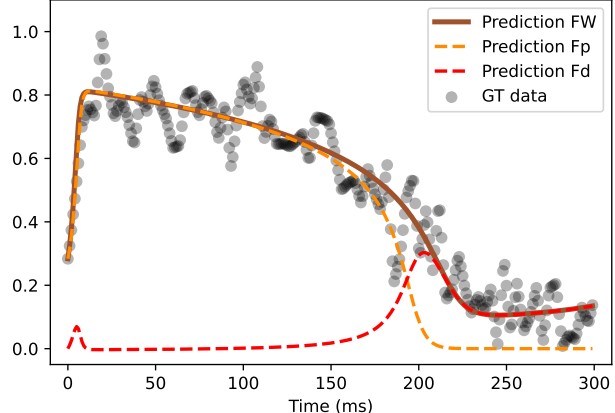

*Figure 3.* Validation results of the framework trained on ROI A data, identified parameters: $\tau_{\text{in}} = 0,613$, $\tau_{\text{out}} = 4, 1$, $\tau_{\text{close}} = 284$. Ground truth (GT) data, prediction of the framework (Prediction FW), decomposition of prediction on physical ($F_p$) and DL ($F_{dl}$) parts.

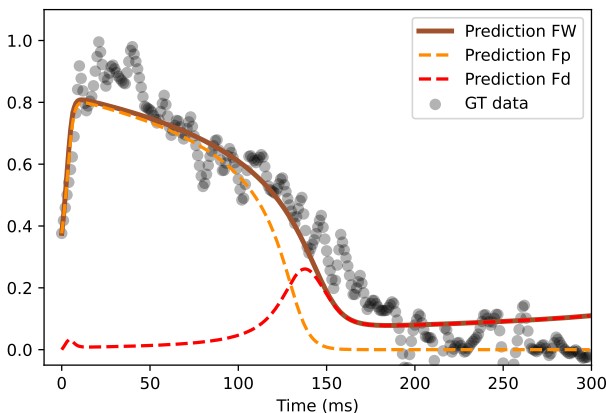

*Figure 4.* Validation results of the framework trained on ROI B data, identified parameters: $\tau_{\text{in}} = 0.745$, $\tau_{\text{out}} = 5$, $\tau_{\text{close}} = 183$. Ground truth (GT) data, prediction of the framework (Prediction FW), decomposition of prediction on physical ($F_p$) and DL ($F_{dl}$) parts.

*framework's name* framework achieved forecasting the dynamics with good accuracy for new data samples from the validation dataset. Furthermore, our framework clearly outperformed the physical model for every dataset, while the contribution of $F_d$ component was still minimal. Despite having a good results on ROI B, the pure data-driven model encountered difficulties to learn the dynamics from ROI A data.

*Table 1.* Mean-squared error, MSE (x $10^{-3}$) of the normalised transmembrane potential (adimensional) forecasting (forecasting horizon of 300 ms). Baseline models: the Physical model (2) and a fully data-driven model (EP-Net 2.0 (Kashtanova et al., 2021)) trained on the same dataset as *Our framework's name*.

| DATASET | METHOD | TRAINING DATA | VALIDATION DATA |
|---------|--------|---------------|-----------------|
| ROI A | *Our framework's name* FRAMEWORK WITH RESNET ($\|F_d\|^2$) | 9.12 (0.16) | 5.72 (0.08) |
| | *Our framework's name* FRAMEWORK WITH MLP ($\|F_d\|^2$) | 9 (0.0785) | 5.37 (0.077) |
| | PHYSICAL MODEL | 14 | 10 |
| | DATA-DRIVEN MODEL | 20 | 9.78 |
| ROI B | *Our framework's name* FRAMEWORK WITH RESNET ($\|F_d\|^2$) | 10 (0.08 ) | 8 (0.07) |
| | *Our framework's name* FRAMEWORK WITH MLP ($\|F_d\|^2$) | 8.79 (0.18) | 7 (0.21) |
| | PHYSICAL MODEL | 14.5 | 9.3 |
| | DATA-DRIVEN MODEL | 7.78 | 6.79 |

## 4. Conclusion

In this article we successfully demonstrated the ability of a novel *Our framework's name* framework to learn real cardiac EP dynamics from data. The main advantage of our proposed framework is its coupled architecture, which allowed us to use a simplified low-fidelity EP model as a physical component of the framework, along with a neural network as a data-driven correction mechanism for the EP model. Our original framework opens up several possibilities in order to introduce prior knowledge in deep learning approaches through explicit equations, as well as to correct the physical model errors from assimilated data.

This combined physics-based data-driven approach may improve cardiac electrophysiology modeling by providing a robust biophysical tool for predictions.

## Acknowledgments

This work has been supported by the French government, through the 3IA Côte d'Azur Investments in the Future project managed by the National Research Agency (ANR) ANR-19-P3IA-0002, the "Research and Teaching chairs in artificial intelligence (AI Chairs) DL4Clim project ANR-19-CHIA-0018, the DeepNum project ANR-21-CE23-0017. The authors are grateful to the OPAL infrastructure from Université Côte d'Azur for providing resources and support.

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
