# OpenReview forum: "Deep Learning Approach for Cardiac Electrophysiology Model Correction"
_ICML.cc/2023/Workshop/IMLH — IMLH 2023 PosterShortPaper_

### Official Review · Reviewer_AXqh · 2023-06-10
**This work presents an innovative framework aimed at learning complex cardiac electrophysiology dynamics using a combination of a simplified low-fidelity physical model and a deep learning approach.**

**Rating:** 5
**Confidence:** 4

**Review:**

Pros:

The method integrates a simplified low-fidelity physical model for cardiac electrophysiology, which accounts for some known physiological behaviors. This component brings physiological interpretability to the framework and could help to improve generalization performance.


Cons:


Simplification of Physical Model: The use of a simplified low-fidelity model could limit the ability of the framework to fully capture complex physiological phenomena. Even though the deep learning component is designed to compensate for this, it may not be capable of completely addressing this limitation.

Generalizability: While the framework demonstrates potential, its ability to generalize across different cardiac diseases and conditions remains uncertain. Further research would be needed to ascertain its broad applicability.

---

### Official Review · Reviewer_BMTs · 2023-06-15
**An interesting work with clinical relevance**

**Rating:** 7
**Confidence:** 3

**Review:**

This work presents a computational framework that leverage complementary physics-based model and machine learning model for cardiac electrophysiology modeling. The proposed framework can handle partial observation.

Pros:
The application is of sufficient clinical relevance.
The methodology is neat. The idea of complementing classical physics-based computation model and deep learning model is technically sound.
Ex vivo data are collected for validating the proposed framework.

Cons:
Given that the proposed method requires iterative optimization, the computational costs need to be discussed in more details.
The experiments are performed in quite a small scale with ex vivo data. The authors are expected to postulate how the model may perform on datasets with larger scale.

---

### Meta-Review · Area_Chair_qemB · 2023-06-20

**Recommendation:** Accept (Poster)
**Confidence:** 5

**Metareview:**

The paper presented a concise and solid study for evaluating deep learning methods for Cardiac electrophysiology model correction. This is an interesting study in the medical intervention domain. The review highlighted concerns regarding the use of a simplified low-fidelity model, which may restrict the framework's capability to comprehensively capture complex physiological phenomena. It is recommended that the authors address this question in further detail within the paper. Overall, this concise case study on cardiac electrophysiology model correction holds significant potential for contributing to the community.

---

### Decision · Program_Chairs · 2023-06-20

Accept (Poster Short Paper)